# Predicting Carotid Artery Stenosis Progression: A Comprehensive Machine Learning Approach

Panagiotis K. Siogkas
*Dept. of Materials Science and Engineering*
*University of Ioannina*
Ioannina, Greece
psiogkas@uoi.gr

Dimitrios Pleouras
*Dept. of Materials Science and Engineering*
*University of Ioannina*
Ioannina, Greece
dipleouras@gmail.com

Vassilis Tsakanikas
*Dept. of Materials Science and Engineering*
*University of Ioannina*
Ioannina, Greece
vasilistsakanikas@gmail.com

Vassiliki Potsika
*Dept. of Materials Science and Engineering*
*University of Ioannina*
Ioannina, Greece
vpotsika@gmail.com

Fragkiska Sigala
*First Propedeutic Department of Surgery*
*National and Kapodistrian University of Athens*
Athens, Greece
drfsigala@yahoo.gr

Dimitrios I. Fotiadis
*Dept. of Materials Science and Engineering*
*University of Ioannina*
Ioannina, Greece
fotiadis@uoi.gr

*Abstract*— Cardiovascular diseases remain a leading global health burden, with carotid artery stenosis progression being a critical determinant of adverse cerebrovascular outcomes. This study aims to enhance the predictive accuracy of carotid stenosis progression by leveraging advanced machine learning algorithms, thereby advancing precision medicine in vascular care. A comprehensive evaluation of six classification models was conducted, including Extreme Gradient Boosting (XGBoost), Support Vector Machine (SVM), Random Forest, Gradient Boosting, Logistic Regression, and Decision Tree. Each model was trained on a curated set of hemodynamic, imaging, and clinical features, and assessed using a 70%–30% stratified train-validation split. The study emphasizes the integration of high-fidelity simulation outputs and ultrasound-derived features in constructing predictive models. Among the classifiers tested, XGBoost demonstrated the highest performance, achieving an AUC of 0.741, accuracy of 70.5%, precision of 70%, and specificity of 88.9% on the validation dataset. Additionally, a 5-fold cross-validation strategy was performed to assess generalizability, with XGBoost achieving a mean AUC of 0.732 and improved overall robustness across folds. These metrics underline its superior capacity to distinguish between stable and progressive stenosis cases. SVM and Gradient Boosting also yielded competitive results, while simpler models lagged in performance. The findings underscore the value of machine learning—particularly ensemble-based approaches such as XGBoost—in predicting stenosis progression. By incorporating rich hemodynamic data and patient-specific imaging features, the model offers a viable tool for early intervention planning. Future work should focus on longitudinal datasets and further validation to support clinical translation and personalized therapeutic strategies.

*Keywords*— *Machine Learning in Carotid Artery Disease, Stenosis Progression, Prognosis*

## I. INTRODUCTION

Carotid artery disease, primarily caused by atherosclerosis, poses a significant public health concern due to its direct association with ischemic strokes—the second leading cause of death globally and a major source of long-term disability [1]. The condition involves the narrowing of the carotid arteries, which are essential vessels supplying oxygenated blood to the brain. As the lumen narrows due to plaque buildup, the risk of stroke increases, particularly when stenosis exceeds clinically significant thresholds. To better understand the mechanisms driving plaque development and progression, researchers have turned to computational fluid dynamics (CFD). CFD provides detailed insight into the hemodynamic environment of the carotid bifurcation, enabling the assessment of parameters such as Wall Shear Stress (WSS), Oscillatory Shear Index (OSI), and pressure distributions, all of which influence plaque vulnerability. Pioneering studies by Hyun et al. [2] and Filipovic et al. [3] highlighted the role of WSS in identifying disturbed flow regions and modeling LDL accumulation, while Marshall et al. [4] and Morbiducci et al. [6] emphasized the necessity of realistic inflow/outflow boundary conditions for accurate modeling. Further refinements include studies by Lee et al. [5], Sousa et al. [7], and Dong et al. [8], who evaluated the effects of stenosis and boundary modeling strategies. Wong et al. [9], Massai et al. [10], and Aristokleous et al. [11] demonstrated the role of vessel geometry and helical flow in shaping hemodynamic stresses. Additional parameters influencing carotid flow, such as external magnetic fields [12], stent design [13], and blood viscosity models [14], were also explored, while Xu et al. [15] and Dai et al. [16] addressed boundary condition formulation and post-treatment flow changes, respectively. In parallel, the field has shifted toward predictive modeling and personalized medicine. Greco et al. [17] and de Weerd et al. [18] employed statistical models to identify individuals at risk of critical stenosis, while Hao-wen Li et al. [19] utilized high-resolution MRI to distinguish between vulnerable and stable plaques. Other studies, including those by Hameed et al. [20], Jodko et al. [21], Raptis et al. [22], and Rezazadeh et al. [23], examined dynamic vessel-wall interactions and vascular remodeling, emphasizing the need for personalized treatment strategies.

Despite these advances, integrating simulation-based hemodynamic features with clinical and imaging data into a single predictive framework remains limited. Our study addresses this gap by combining patient-specific CFD-derived metrics, ultrasound-based stenosis measures, and clinical risk factors to train and evaluate machine learning models for stenosis progression prediction. By comparing multiple classifiers and identifying the most robust model, we aim to enhance early risk stratification and support the transition toward precision vascular healthcare.

## II. METHODS

### A. Dataset

This study utilized a dataset of 87 carotid arteries collected in a prospective clinical setting. Inclusion criteria required the presence of non-trivial atherosclerotic plaque at baseline, availability of high-resolution MRI suitable for 3D lumen reconstruction, and complete one-year follow-up imaging using duplex ultrasound. Cases with motion artifacts, segmentation failure, or missing clinical data were excluded. MRI scans were acquired using a 1.5-Tesla GE Signa HDx system (GE Healthcare, Waukesha, WI, USA) with a bilateral four-channel carotid coil (Machnet BV, Eelde, the Netherlands). 3D lumen reconstruction was based on Time-of-Flight (ToF) sequences. Ultrasound imaging was used to determine peak systolic and end-diastolic velocities for boundary conditions and to assess disease progression at follow-up. All participants provided informed written consent before imaging. The study protocol received approval from the relevant institutional ethics committees, and all procedures were conducted in accordance with ethical guidelines for research involving human subjects.

### B. 3D Reconstruction

In the context of the present study, only the lumen was required to be reconstructed in 3D for the subsequent blood flow simulations. For this purpose, only the Time of Flight (ToF) images were utilized by our in-house developed 3D reconstruction algorithm the process of which can be summarized in the following steps. Region of interest segmentation: The initial phase is to demarcate the critical areas of interest, which encompass the inner arterial border which represents the lumen. For this purpose, a trio of specialized deep learning networks was established. In particular, an ensemble of 485 Time-of-Flight (ToF) images, gathered from 42 distinct sub-jects, was scrupulously annotated by a pair of experienced radiologists, forming the foundation for a training dataset. This compilation of data was then harnessed to fine-tune three distinct UNET architectures, each designed to precisely delineate one of the targeted zones. 3D level set: Post-delineation, an advanced morphological process is executed on the 3D assemblage of the consecutively delineated 2D slices. This step is critical for crafting a three-dimensional surface representation of the delineated regions. 3D meshing: Following this, a sophisticated algorithm known as marching cubes is utilized to transform the 3D surface representation into a mesh, culminating in the construction of the intricate arterial model. This systematic and integrated technique ensures the creation of finely detailed and precise 3D models of the carotid arterial lumen, facilitating subsequent examinations and research into vascular conditions.

## III. BLOOD FLOW MODELING

Transient simulations of blood flow through 3D reconstructed carotid artery geometries were performed using specific boundary conditions from carotid UltraSound (US) screenings. These included flow velocity profiles captured over at least three cardiac cycles for every artery. Blood flow dynamics were modeled with the Navier-Stokes and continuity equations, described as follows:

$$\rho \frac{\partial \boldsymbol{v}}{\partial t} + \rho(\boldsymbol{v} \bullet \nabla)\boldsymbol{v} - \nabla \bullet \boldsymbol{\tau} = 0, \qquad (1)$$

$$\nabla \bullet (\rho \boldsymbol{v}) = 0, \qquad (2)$$

where '$\boldsymbol{v}$' is the velocity vector of the blood and '$\boldsymbol{\tau}$' the stress tensor, which itself is defined by:

$$\nabla \bullet (\rho \boldsymbol{v}) = 0, \qquad (3)$$

$$\varepsilon_{ij} = \frac{1}{2}(\nabla \boldsymbol{v} + \nabla \boldsymbol{v}^T), \qquad (4)$$

with $\delta_{ij}$ as the Kronecker delta, μ the dynamic viscosity of the blood, p the blood pressure, and $\varepsilon_{ij}$ the strain tensor. Assuming Newtonian fluid characteristics for blood with a density of 1050 kg/m³ and a dynamic viscosity of 0.0035 Pa·s, simulations were run on ANSYS® v16.2, employing a mesh exclusively composed of tetrahedra with a maximum element size of 0.16 mm. Mesh size and simulation quality were optimized through sensitivity analysis, adhering to a convergence criterion of $10^{-4}$ and limiting iterations to 150 per time step. To accurately simulate patient-specific blood flow dynamics, we employ a detailed approach that involves generating mass flow rate profiles from Carotid Ultrasonography (US) images. These customized profiles are then applied to the inlet, which corresponds to the beginning of the Common Carotid Artery (CCA), and the outlet of the External Carotid Artery (ECA). This approach is grounded in the precise Ultrasonography data from clinical centers, which captures the flow velocity profiles for at least two out of the three main arterial branches: CCA, Internal Carotid Artery (ICA), and/or ECA. Each US image acts as a detailed source of velocity measurements and waveforms for several cardiac cycles across different arteries. A patient-specific flow velocity profile is constructed from Peak Systolic Velocity (PSV) and End Diastolic Velocity (EDV) values using a specialized MATLAB script developed in-house. Moreover, specific durations of cardiac cycles are determined from pulse rate data, adding temporal dimensions to the simulations. This holistic approach not only utilizes the depth of the US data but also integrates it precisely to ensure high fidelity in replicating the unique aspects of patient-specific blood flow dynamics. The velocity profiles are subsequently converted into mass flow rate profiles, as illustrated in Eq. (5).

$$\dot{m} = \rho V A \qquad (5)$$

$\dot{m}$ signifies the mass flow rate, $\rho$ represents the density of blood, $V$ indicates the velocity of blood, and $A$ is the cross-sectional area where the velocity is measured. For the Internal Carotid Artery (ICA), we apply a boundary condition of zero pressure, with a comprehensive justification for this decision detailed in the results section. Furthermore, a no-slip boundary condition is implemented along the arterial walls to ensure the velocity there is zero, accompanied by a no-penetration condition that prevents any fluid from crossing the boundary.

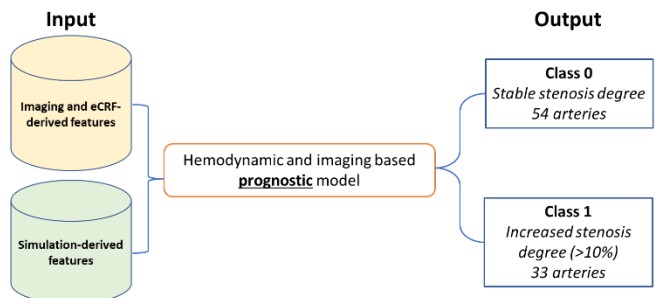

*Figure 1: Problem definition for the plaque progression model.*

## IV. RESULTS

The development of the final plaque progression model treats the advancement of carotid artery disease as a complex multivariate binary classification issue, focusing on stenosis measurements obtained from detailed ultrasound (US) screenings of patients. Figure 1 graphically displays this classification approach, visually differentiating between arteries with stable or slight stenosis and those with notable stenosis progression, underscoring the model's role in effectively managing carotid artery health and identifying risk factors.

### A. Model Input Features

In this study, a total of eighteen input parameters were used to train the predictive models, drawing from three main categories: blood flow simulation outputs, ultrasound measurements, and clinical data recorded in the electronic Case Report Form (eCRF). From the simulation results, key hemodynamic indicators were extracted, including the Time-Averaged Endothelial Shear Stress (TAESS), Oscillatory Shear Index (OSI), and pressure ratios between branches of the carotid artery ($P_{ECA}/P_{CCA}$ and $P_{ICA}/P_{CCA}$). The proportion of the vessel area exposed to low TAESS and high OSI was also calculated and normalized, capturing regions with potential susceptibility to atherosclerotic changes (Figure 2).

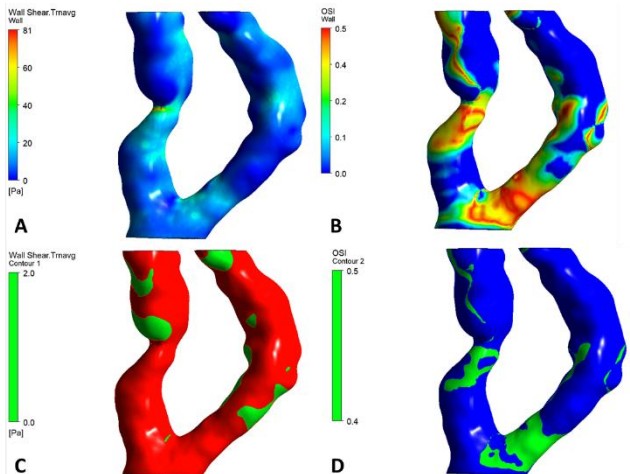

*Figure 2: A) TAWSS distribution, B) OSI distribution, C) Areas of low TAWSS depicted in green, D) Areas of high OSI depicted in green.*

Complementing these were patient-specific ultrasound-derived metrics, such as the degree of stenosis in the internal and external carotid arteries and the peak systolic velocity (PSV). Finally, the inclusion of clinical variables—such as smoking status, the presence of diabetes or hypertension, body mass index (BMI), alcohol consumption, and statin use—allowed for the integration of relevant cardiovascular risk factors. This combined approach ensured that the model accounted for both anatomical and physiological characteristics relevant to plaque progression. Table 1 presents the features that are used within the ML-model approach.

*Table 1: Utilized features by the prognostic ML model.*

| Feature type | Feature |
|---|---|
| **Imaging** | Artery, Mean arterial pressure |
| **Simulation-based** | Peak TAESS, $P_{ECA}/P_{CCA}$, $P_{ICA}/P_{CCA}$, Vessel average TAESS, Vessel average OSI, Area of low TAESS, Area of high OSI, Normalized area of low TAESS, Normalized area of high OSI |
| **eCRF-based** | Smoking, diabetes, hypertension |

### B. Model Selection and Evaluation

Multiple classifiers were evaluated for their ability to predict stenosis progression, including Logistic Regression, Random Forest, Gradient Boosting, Support Vector Machine (SVM), Decision Tree, and Extreme Gradient Boosting (XGBoost). Arteries were classified into two groups based on follow-up imaging: Class 0 (54 arteries) for stable or minimally changing stenosis (<10% increase) and Class 1 (33 arteries) for cases with a stenosis increase greater than 10%. This threshold was chosen in consultation with clinical collaborators to reflect changes considered clinically relevant while reducing the impact of measurement variability. The decision to set a 10% threshold for significant stenosis changes was made to capture clinically meaningful alterations. Increases below this threshold, such as 5%, may lack statistical relevance and could result from imaging inaccuracies or minor measurement variances. In establishing this threshold, the model benefits from the clinical insights of medical professionals, enhancing its accuracy and applicability by relying on expertly interpreted ultrasound data. A stratified 70-30 split was used for training and validation, ensuring class balance across both sets. Each classifier was trained using the same set of simulation-based and clinical features. The models were evaluated based on a range of metrics: Area Under the Curve (AUC), Accuracy, Precision, and Specificity. These metrics provide a comprehensive overview of model performance, assessing both sensitivity to true positives and resistance to false positives.

*Table 2: Performance comparison of classifiers on the validation set. The best-performing classifier is shown in green.*

| Classifier | AUC | Accuracy | Precision | Specificity |
|---|---|---|---|---|
| Logistic Regression | 0.703 | 0.682 | 0.667 | 0.778 |
| Random Forest | 0.712 | 0.682 | 0.640 | 0.833 |
| Gradient Boosting | 0.723 | 0.682 | 0.650 | 0.833 |
| Support Vector Machine | 0.709 | 0.682 | 0.654 | 0.833 |
| Decision Tree | 0.663 | 0.636 | 0.579 | 0.778 |
| **XGBoost** | **0.741** | **0.705** | **0.700** | **0.889** |

The best-performing model was the **XGBoost classifier**, achieving an AUC of 0.741, an accuracy of 70.5%, a precision of 70.0%, and a specificity of 88.9% (Table 2). This superior performance highlights the model's ability to balance sensitivity and specificity while effectively distinguishing between stable and progressive stenosis cases. To assess the individual contribution of each feature modality, we conducted an ablation analysis using the XGBoost classifier trained separately on (i) simulation-based hemodynamic features, (ii) clinical variables, and (iii) imaging-derived parameters. Performance metrics across these single-modality models were then compared to the multimodal configuration that included all features. As shown in Table 3, the multimodal model achieved the highest

predictive performance, with an AUC of 0.741 and specificity of 88.9%. Notably, the CFD-only model outperformed both the clinical-only and imaging-only configurations, indicating that hemodynamic factors hold considerable predictive value on their own. However, integrating all feature domains led to substantial gains across accuracy, precision, and specificity, confirming the complementary nature of these modalities and validating the added value of multimodal integration. **To further assess generalizability, we applied a 5-fold stratified cross-validation strategy using the balanced dataset.** XGBoost maintained its superior performance, achieving a mean AUC of **0.732 ± 0.047**, accuracy of **69.6% ± 4.5%**, precision of **67.8% ± 6.2%**, recall of **64.5% ± 5.7%**, and specificity of **73.5% ± 7.1%**. These results (Table 5) reinforce the robustness of XGBoost and confirm its capacity to generalize across folds without overfitting.

*Table 3: Ablation analysis using the XGBoost classifier trained separately on (i) simulation-based hemodynamic features, (ii) clinical variables, and (iii) imaging-derived parameters and compared to the Multimodal analysis.*

| Feature Set | AUC | Accuracy | Precision | Specificity |
|---|---|---|---|---|
| CFD-only | 0.661 | 63.6% | 63.6% | 77.8% |
| Clinical-only | 0.625 | 63.6% | 61.5% | 66.7% |
| Imaging-only | 0.538 | 59.1% | 50.0% | 66.7% |
| **Multimodal** | **0.741** | **70.5%** | **70.0%** | **88.9%** |

A SHAP analysis was also performed to identify the features with greater importance on the produced results. The most influential feature in predicting stenosis progression was the normalized area with low TAWSS, aligning with established literature that links disturbed flow regions to atherogenesis [24]-[27]. Other critical features include absolute and average TAWSS, and regions with high oscillatory shear index (OSI), reflecting their role in endothelial dysfunction and plaque instability (Table 4).

*Table 4: Top five ranked features for the XGBoost classifier, based on SHAP importance scores.*

| Rank | Feature | SHAP Importance |
|---|---|---|
| 1 | Area of low TAWSS/Total vessel area (%) | 0.232 |
| 2 | Area of low TAWSS (m²) | 0.198 |
| 3 | Vessel average TAWSS (Pa) | 0.154 |
| 4 | Area of high OSI/Total vessel area (%) | 0.129 |
| 5 | Vessel average OSI | 0.101 |

Figure 3 illustrates the confusion matrix for the XGBoost classifier. The model achieved 9 true positives and 11 true negatives, indicating a strong ability to correctly identify both cases of stenosis progression and stable arteries. Importantly, there was only 1 false negative, suggesting minimal risk of missing clinically significant progression. While 6 false positives were observed—reflecting some tendency to over-predict progression—the overall balance of the matrix supports the model's reliability. In addition to standard metrics, the model achieved a true positive rate (TPR) of 90.0%, true negative rate (TNR) of 64.7%, false positive rate

(FPR) of 35.3%, and false negative rate (FNR) of 10.0%, further supporting its clinical interpretability. These results reinforce XGBoost's strong performance across all key metrics and highlight its clinical relevance in early risk stratification for carotid artery disease.

*Table 5: Performance of the XGBoost model using 5-fold stratified cross-validation. Values are reported as mean ± SD across folds.*

| Metric | Mean ± SD |
|---|---|
| **AUC** | 0.732 ± 0.047 |
| **Accuracy** | 69.6% ± 4.5% |
| **Precision** | 67.8% ± 6.2% |
| **Recall** | 64.5% ± 5.7% |
| **Specificity** | 73.5% ± 7.1% |

All machine learning models were trained using default hyperparameter settings as provided by the respective implementations. This decision was made to ensure consistency across classifiers and to focus the analysis on feature modality comparisons rather than on model-specific tuning. Future work will include systematic hyperparameter optimization using grid search or Bayesian strategies.

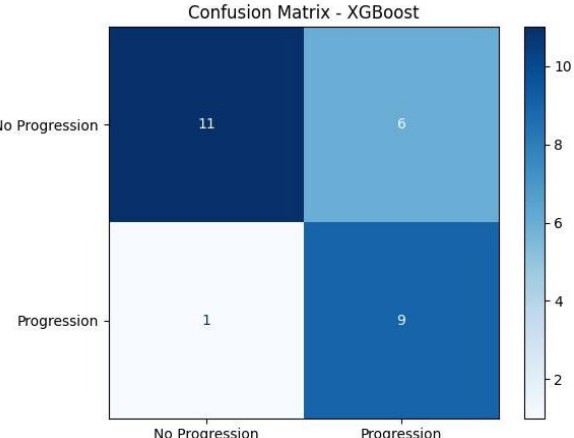

*Figure 3: Confusion matrix for the dominant classifier (XGBoost) on the validation dataset (30%).*

## V. DISCUSSION

Previous research has extensively explored blood flow modeling in the carotid bifurcation, particularly focusing on identifying regions of altered Wall Shear Stress (WSS), applying boundary conditions using lumped parameter models, and examining the influence of arterial lesions on hemodynamic indices. While these approaches have provided important insights, the present study expands the scope significantly by leveraging patient-specific blood flow simulations to derive a rich set of hemodynamic parameters. These features, when combined with ultrasound-derived metrics—such as peak systolic velocity and stenosis severity—form the foundation of a robust prognostic model for predicting plaque progression. Our method integrates high-fidelity imaging and simulation data to reconstruct the carotid artery lumen in 3D and evaluate the mechanical environment under realistic physiological conditions. Key parameters such as Time-averaged Endothelial Shear Stress (TAESS), Oscillatory Shear Index (OSI), and pressure ratios ($P_{ECA}/P_{CCA}$, $P_{ICA}/P_{CCA}$) were extracted and combined with relevant clinical data from the electronic Case Report Form (eCRF), including smoking status, diabetes, and hypertension. This multimodal approach offers a comprehensive

characterization of plaque vulnerability and progression risk. To assess predictive performance, six machine learning classifiers were trained and tested on this dataset: Logistic Regression, Random Forest, Gradient Boosting, Support Vector Machine (SVM), Decision Tree, and Extreme Gradient Boosting (XGBoost). The results revealed a clear stratification in model effectiveness. Notably, XGBoost emerged as the top performer, achieving the highest values across all evaluation metrics: AUC, accuracy, precision, and specificity. Its ability to handle heterogeneous feature sets and model complex non-linear relationships likely contributed to this strong performance, positioning it as a promising tool for clinical decision support in managing carotid artery disease. SVM and Gradient Boosting also demonstrated competitive performance, offering high specificity and balanced classification power. These models may serve as viable alternatives in settings where interpretability or computational efficiency is prioritized. In contrast, simpler models like Decision Tree and Logistic Regression underperformed, suggesting that more advanced ensemble-based or margin-based approaches are better suited for capturing the intricate patterns present in plaque progression. A rigorous 70-30 train-validation split with stratified sampling ensured that class balance was preserved, while maintaining the independence of the validation set. Complementarily, the use of 5-fold cross-validation confirmed the consistency and generalizability of XGBoost's performance, reducing the likelihood of overfitting due to sample stratification and further supporting the robustness of the findings. This strategy, combined with internal performance metrics such as AUC, strengthens the credibility of our findings and supports the potential for model generalizability. Despite the encouraging results, certain limitations warrant consideration. The dataset represents a single snapshot in time, limiting our ability to model dynamic plaque evolution or capture temporal variability in hemodynamic factors. Additionally, although the dataset was sufficiently large to support exploratory modeling, broader generalizability to diverse patient populations may require expanded cohorts with greater demographic and anatomical variability. Finally, the accuracy of the simulations is inherently tied to the fidelity of the vascular reconstructions and the quality of ultrasound imaging—both of which can introduce variability in the boundary conditions and potentially affect model predictions. Although the validation strategy maintained independence between training and test sets, the reliance on a single time point and a relatively small sample size introduces a risk of overfitting. Additionally, the use of default hyperparameters may have limited the performance of certain classifiers. Future work will incorporate model-specific hyperparameter optimization to enhance predictive accuracy and robustness. We also plan to expand the dataset with multi-institutional and longitudinal cases to improve generalizability. Finally, repeated cross-validation and bootstrapped confidence intervals will be applied to provide a more rigorous assessment of performance variability.

## VI. CONCLUSIONS

This study demonstrates the feasibility and potential utility of integrating simulation-based hemodynamic features, ultrasound imaging, and clinical data into a machine learning framework for predicting carotid stenosis progression. Among the evaluated models, XGBoost outperformed other classifiers, offering favorable accuracy, precision, and specificity metrics. The results suggest that patient-specific CFD parameters contribute meaningfully to risk stratification and may complement existing clinical indicators. While these findings are promising, we acknowledge that the reported performance levels—though competitive—do not yet support clinical deployment. Further work is needed to validate the model on larger, multi-center, and longitudinal datasets, as well as to assess its robustness under varying imaging conditions and clinical settings. These efforts will be essential for refining the model and translating it into a reliable clinical decision-support tool.

ACKNOWLEDGMENT

This work has received funding from the European Union's Horizon 2020 research and innovation programme under grant agreement No 755320, as part of the TAXINOMISIS project.

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
