# OpenReview forum: "Predicting Carotid Artery Stenosis Progression: A Comprehensive Machine Learning Approach"
_IEEE.org/EMBS/BHI/2025/Conference — BHI 2025_

### Official Review · Reviewer_v8nu · 2025-06-27
**Predicting Carotid Artery Stenosis Progression: A Comprehensive Machine Learning Approach**

**Confidence:** 4
**Clarity Of Writing:** good
**Clinical Significance:** good
**Methodological Novelty:** fair
**Overall Rating:** 6
**Final Rating:** 7

**Experiments And Results:**

fair

**Questions For The Authors:**

Can you clarify the dataset size, class distribution, and any steps taken to address class imbalance? This will help assess the robustness of the reported performance.

Why was a single 70-30 split used instead of k-fold cross-validation? Would the results be stable under different folds?

Have you considered interpretability methods (e.g., SHAP, LIME, or feature importance from tree models)? This would be valuable for clinician adoption.

Are the ultrasound-derived features standardized across imaging systems or operators? This impacts generalizability.

Do you plan to validate the models on a prospective or external dataset in future work? This would significantly strengthen the clinical utility of your approach.

**Strengths:**

The integration of multimodal data (hemodynamic, imaging, and clinical) enriches the feature space and reflects real-world complexity.

The use of multiple models, with comparative analysis, offers transparency in model selection.

The emphasis on ensemble methods, particularly XGBoost, aligns with best practices in predictive modeling for structured health data.

Clinical relevance is clearly articulated, aiming to aid early intervention strategies in vascular care.

The metrics used (AUC, accuracy, precision, specificity) are appropriate and clearly reported.

**Summary Of The Paper:**

This study presents a machine learning-based framework for predicting the progression of carotid artery stenosis using a combination of hemodynamic, imaging, and clinical features. Six classifiers were evaluated: XGBoost, SVM, Random Forest, Gradient Boosting, Logistic Regression, and Decision Tree. Models were trained using a 70–30 stratified train-validation split. The integration of high-fidelity simulation and ultrasound-derived features was emphasized. XGBoost achieved the best performance, with an AUC of 0.741, accuracy of 70.5%, precision of 70%, and specificity of 88.9%. The paper suggests ensemble methods are particularly suitable for this problem and recommends future validation on longitudinal datasets.

**Weaknesses:**

The dataset size and specific patient cohort characteristics (e.g., number of progressive vs. stable cases) are not clearly described—this is essential for assessing generalizability.

The methodology for feature selection, preprocessing, and missing data handling is not detailed.

The evaluation relies solely on a single random stratified split (70-30); the results would be more robust with k-fold cross-validation.

While the models are well-selected, interpretability (e.g., SHAP values or feature importance) is not discussed, which is crucial in clinical settings.

No external validation dataset is used, limiting the clinical applicability at this stage.

---

### Official Review · Reviewer_7jhW · 2025-07-06
**Comprehensive Evaluation of Machine Learning for Carotid Stenosis Progression Prediction**

**Confidence:** 3
**Clarity Of Writing:** good
**Clinical Significance:** great
**Methodological Novelty:** good
**Overall Rating:** 6
**Final Rating:** 6

**Experiments And Results:**

good

**Questions For The Authors:**

1. Please discuss the risk of overfitting and address the model’s ability to generalize. Do you have plans for validation on external or longitudinal datasets?

2 .Did you assess potential redundancy or multicollinearity among the features? Consider reporting whether feature selection or dimensionality reduction techniques were explored.


3.It is recommended to use cross-validation or bootstrapping to provide confidence intervals for performance metrics.


4. Please provide more information about the imaging protocols and data acquisition required for practical clinical use.


5 .Please clarify how hyperparameters for each machine learning model were selected or tuned.

**Strengths:**

1. Comprehensive Feature Set: The study integrates CFD-based hemodynamics, ultrasound imaging, and electronic clinical records, providing a rich multimodal dataset.

2. Comparative Analysis: Multiple state-of-the-art machine learning models are compared on the same task, and metrics such as AUC, accuracy, precision, and specificity are reported.

3. Explainability: SHAP analysis identifies the most important features influencing prediction.

4. Clinical Relevance: The findings are directly relevant to early risk stratification and planning of intervention in vascular care.

**Summary Of The Paper:**

This paper presents a thorough machine learning pipeline for predicting the progression of carotid artery stenosis using a combination of simulation-based hemodynamic parameters, ultrasound imaging features, and clinical data. The authors benchmark six different classifiers (XGBoost, SVM, Random Forest, Gradient Boosting, Logistic Regression, Decision Tree) and find that XGBoost achieves the best performance. Feature importance analysis via SHAP is provided, and the results highlight the utility of patient-specific biomechanical and imaging data for risk prediction and early intervention.

**Weaknesses:**

1. Although the sample size (87 arteries) is reasonable, the study only analyzes a single time point, which may limit generalizability and risks overfitting.
2. The manuscript does not clarify whether there is redundancy or multicollinearity among the input features.
3. The validation strategy is limited to stratified train-test splits, without the use of cross-validation or bootstrapping for robust performance estimation.
4.  Details on the choice or tuning of machine learning model hyperparameters are not provided.

---

### Official Review · Reviewer_cN4s · 2025-07-07
**Review of a Machine Learning-Based Approach to CAS Progression Prediction: Innovative but Preliminary**

**Confidence:** 3
**Clarity Of Writing:** good
**Clinical Significance:** excellent
**Methodological Novelty:** fair
**Overall Rating:** 5

**Experiments And Results:**

good

**Questions For The Authors:**

1. The manuscript states that “this study demonstrates the feasibility and effectiveness … for predicting carotid stenosis progression.”. How the term “effectiveness” is justified? Reported metrics such as accuracy and precision around 70% may indicate preliminary promise, but they are likely insufficient for clinical deployment.

**Strengths:**

1. This manualscript is written in a clear structure, and well-defined and clinical important motivation.
2. The methods are also described in a clear way. Large amounts of previous literatures are present for introducting this task.
3. The problem is clear defined.
4. Different models are used to investigate the machine learning-based prediction of carotid artery stenosis.
5. As a part of the method——“Integrating simulation-based hemodynamic features for predicting carotid artery stenosis", is interesting.

**Summary Of The Paper:**

This work focusing on an clinical important theme that predict carotid artery stenosis (CAS) with method learning-based methods. The integration of simulation-based features for the machine learning model is innovative. The results is decent, and the overall structure of the manuscript is clear.

As a preliminary investigation of the utilization of simulation-based features in machine learning for carotid artery stenosis, it remains many topics need to be investigate further, such as compare with similar works methods, evaluate the methods (utilization of simulation-based features) in similar tasks, among others.

**Weaknesses:**

1. While this paper addresses a clinically meaningful decision problem in carotid artery stenosis (CAS), it remains unclear what the key novel contributions are. A number of high-performing machine learning models for CAS prediction or CAS related tasks (e.g., hemodynamic prediction) already exist in the literature.
For example:
- Wang et al. (2022) [doi: 10.3389/fphys.2022.1094743]
- Liang et al. (2021) [doi: 10.1007/s10237-021-01497-7]
- Diao et al. (2021) [arXiv:2108.00296]

2. The investigation appears to be in an early stage and would benefit from additional experimentation and analysis.

Specifically:

- The model's performance should be further improved and rigorously evaluated.

- Comparative analysis with state-of-the-art models should be included to contextualize the results.

- Evaluation on related tasks or external datasets could strengthen the claim of generalizability and robustness.

Despite the aforementioned limitations, the study demonstrates the potential of the proposed method in addressing CAS prediction problems. The direction is clinically relevant, and the integration of machine learning into this domain remains a promising area of research. With further validation, performance improvement, and more comprehensive comparisons to existing work, this method could contribute meaningfully to decision support systems in vascular health.

**Suggestions**:

1. Consider placing figures at the top of the pages where they are referenced. This improves the visual flow of the manuscript and makes it easier for readers to follow the discussion alongside the relevant visuals.

2. Convert all raster images (e.g., PNG, JPEG) into vector formats (e.g., SVG, PDF, EPS) where applicable. This will ensure better clarity and scalability, especially when readers zoom in, which is important for examining detailed visual elements in scientific figures.

3. In addition to standard metrics, please consider also including True Positive Rate (TPR), False Positive Rate (FPR), True Negative Rate (TNR), and False Negative Rate (FNR) in the classification evaluation. These metrics provide more granular insights into model performance and are particularly valuable for clinical interpretation and application.

4. Please remove the use of quote symbols of formula in the main text.

5. The first severl sentences (The development of the final plaque .... minor measurement variances.") the result part are describing evaluation methods. Please consider move it into the method part, or a single section like "evaluation", or present in other forms.

6. Section “II. MATERIALS AND METHODS” -> “METHODS”

7. "eighty-seven (87) " -> "87" in Section II.A's the first sentence.

8.  "recon-structed" -> "reconstructed" in Section II.B's the first sentence.

9. "The symbol \dot m" -> "\dot m" in Section III.

---

### Official Review · Reviewer_SAtK · 2025-07-15
**Review of Predicting Carotid Artery Stenosis Progression: A Comprehensive Machine Learning Approach**

**Confidence:** 4
**Clarity Of Writing:** good
**Clinical Significance:** fair
**Methodological Novelty:** fair
**Overall Rating:** 4
**Final Rating:** 7

**Experiments And Results:**

fair

**Questions For The Authors:**

Would you provide clear details about the data source, curation process, and selection criteria for the 87 carotid arteries? Is this the complete available dataset or a subset, and if the latter, what was the selection rationale?

What specific limitations exist in current stenosis progression prediction methods that your approach addresses? Can you provide concrete examples of where existing methods fail and how your multimodal approach specifically improves upon them?

Can you provide ablation studies comparing single-modality performance (CFD-only, clinical-only, imaging-only) against your multimodal approach? How do you demonstrate that combining features actually improves prediction rather than simply adding complexity?

Was any hyperparameter tuning performed for the models, especially XGBoost? Or if default values are used?

**Strengths:**

The paper combines blood flow simulations, medical images, and patient data in a way that goes beyond traditional risk assessment methods. The use of patient-specific computer simulations with detailed 3D models from MRI scans adds complexity to the analysis. The comparison of six different models using multiple evaluation measures improves reliability, which is important in medical settings where avoiding false predictions matters. The SHAP score helps interpret the feature importance, making the results easier to understand and trust for clinical use. The paper shows a pretty clear explanation in the results.

**Summary Of The Paper:**

The paper applies a machine learning approach to predict carotid artery stenosis progression with the goal of improving prediction of adverse cerebrovascular outcomes. The study integrates hemodynamic features derived from computational fluid dynamics (CFD) simulations with imaging and clinical data to train six different classification models on a dataset of 87 carotid arteries. Using a binary classification framework with >10% stenosis increase as the progression threshold, the paper evaluates model performance using AUC, accuracy, precision, and specificity on a validation set. XGBoost achieved the best performance (AUC 0.741, accuracy 70.5%, specificity 88.9%), highlighting the utility of ensemble models and multimodal feature integration for predicting stenosis progression.

**Weaknesses:**

**The dataset justification is unclear.** The paper mentions a "curated dataset" of 87 carotid arteries without explaining the data source, curation process, or whether this represents the maximum available data or a selected subset. This lack of transparency raises questions about data selection bias and generalizability.

**The gap of knowledge and the study motivation is not clearly articulated.** The paragraph that explains the relevant studies gave the impression that a lot of things are done in the field. Although the paper claims that combining CFD features with clinical data is "limited," they don't explain what exactly is “not enough” with the previous methods or why this combination is necessary. This makes it hard to understand if the study actually solves a real problem.

**The multimodal claims (inclusion of hemodynamic, clinical and imaging features) are not supported by evidence.** The paper only trains the models with all three types of features and no single modality comparison, or any performance comparison with existing methods. The claim of combining the features leads to better performance is not reliable.


In the result, “The most influential feature in predicting stenosis progression was the normalized area with low TAWSS, aligning with established literature that links disturbed flow regions to atherogenesis.” mentioned statement from earlier literature.  but no citation

When the paper concludes simple models perform worse than complex ones, it assumes this is related to the biological complexity, but they should also consider that ensemble methods may handle high dimensional features better. Without comparison to existing stenosis prediction methods, it's unclear whether this approach offers clinically meaningful improvements over current practice. Or circle back to the previous comment, the comparison between small feature space versus large feature space may be needed. The performance analysis feels like applying a standard machine learning model to existing CFD results without clear evidence of advancement in the clinical impact.